# Solidification and Liquation Cracking in Welds of High Entropy CoCrFeNiCu_x_ Alloys

**DOI:** 10.3390/ma16165621

**Published:** 2023-08-14

**Authors:** Ping Yu, Sindo Kou, Chun-Ming Lin

**Affiliations:** 1Department of Materials Science and Engineering, University of Wisconsin, Madison, WI 53706, USA; lyyuping@aliyun.com (P.Y.); kou@engr.wisc.edu (S.K.); 2Department of Mechanical Engineering, Minghsin University of Science and Technology, Hsinchu 30401, Taiwan

**Keywords:** k high entropy alloys, solidification cracking, liquation cracking, gas-tungsten arc welding (GTAW)

## Abstract

High entropy CoCrFeNiCu_x_ alloys with a Cu molar ratio of x ≈ 0, 0.5, 1, 1.5 and 2 were arc welded. Solidification cracking occurred in the fusion zones of alloys with x ≈ 0.5, 1 and 1.5. Cu-rich material was observed around cracks, increasing in quantity with increasing Cu content. Liquation cracking occurred in the partially melted zone next to the fusion zone, and it propagated into the fusion zone as solidification cracking. A recently proposed index for the susceptibility to solidification cracking was tried, i.e., |*dT/d(f_S_*)^1/2^| near *(f_S_*)^1/2^ = 1, where *T* is temperature and *f_S_* the solid fraction. The index was higher in alloys with x ≈ 0.5, 1.0 and 1.5, consistent with the solidification cracking observed.

## 1. Introduction

Yeh et al. [1] and Cantor [2] independently reported high entropy alloys (HEAs) in 2004. HEAs usually contain several elements in essentially equimolar composition. They can be face-centered cubic (fcc), body-centered cubic (bcc) or hexagonal closest packed (hcp) in structure and have good strength, ductility and corrosion resistance. The equimolar CoCrFeNi alloy has been widely used as the base alloy for adding additional alloying elements, e.g., Mn, Al, Cu, Ti, Mn or Mo, to design new HEAs [3]. HEA CoCrFeNiCu, which contains five elements essentially equimolar in composition, has been studied frequently [4,5,6,7,8,9,10,11,12,13]. CoCrFeNiCu_0.5_ has also been studied, which contains essentially Co, Cr, Fe and Ni each at 22.2 at% and Cu at 11.1 at% [14].

The solidification microstructure of HEA CoCrFeNiCu has been studied by melting and solidification in a crucible [5,11]. One exception is the directional solidification of CoCrFeNiCu [8]. The alloy was cast and machined into a rod of 3.9 mm diameter and inserted into an alumina tube (4 mm ID and 6 mm OD). The tube was withdrawn at predetermined speeds from a 1600 °C Bridgman furnace. The effect of the withdrawal speed on the directional solidification microstructure was shown. No quenching during directional solidification was conducted to reveal the solidification microstructure evolution. 

The study on the solidification microstructure of HEA CoCrFeNiCu in welding has focused on fiber laser welding [12,13]. CoCrFeNiCu is likely to have a much wider freezing temperature range than CoCrFeNi and hence a much higher susceptibility to hot cracking. However, hot cracking was avoided in fiber laser welding of CoCrFeNiCu [12,13] perhaps due to the small heat input per unit length of the fiber laser weld.

Kou [15] proposed a simple index for the susceptibility to solidification cracking, that is, |*dT/d(f_S_)*^1/2^| near the grain roots, i.e., near *(f_S_)*^1/2^ = 1, where *T* is temperature and *f_S_* the solid fraction. As illustrated in Figure 1, he showed the grain radius *r_d_* is proportional to *(f_S_*)^1/2^. Thus, for a given temperature drop |*dT*|, a higher |*dT/d(f_S_)*^1/2^| indicates a smaller *d(f_S_)*^1/2^ and |*dr_d_*| and hence a slower lateral growth for grains to bond to each other to resist solidification cracking under tension. The slower lateral growth also allows the narrow channel near the roots to grow longer before bonding, thus slowing down the liquid feeding through the channel that is needed to resist solidification cracking. A convenient option for the index is the maximum |*dT/d(f_S_)*^1/2^| up to *(f_S_)*^1/2^ = 0.99, i.e., *f_S_* = 0.98 [16]. Based on the alloy composition, the *T-f_S_* curve and hence *T-(f_S_)*^1/2^ curve can be plotted using, for example, commercial thermodynamics software Pandat2019 [16] and databases PanHEA2019 [17] of CompuTherm, LLC, Madison, WI. *dT/d(f_S_)*^1/2^ is the slope of the *T-(f_S_)*^1/2^ curve, and |*dT/d(f_S_)*^1/2^|is the steepness. The validity of the index has been verified against the experimental data of Al alloys [18,19,20] and steels [21,22,23].

The purpose of the present study was to explore the arc weldability of CoCrFeNiCu_x_ alloys by examining their susceptibility to hot cracking during welding, including solidification cracking in the fusion zone and liquation cracking in the partially melted zone. The index |*dT/d(f_S_)*^1/2^| for the susceptibility to solidification cracking will be compared with the solidification cracking observed in arc welding.

## 2. Materials and Methods

The preparation of the HEAs CoCrFeNiCu_x_ is described as follows. Raw elemental metals above 99.9 at%, about 200 g total, were arc-melted in a water-cooled Cu mold inside a vacuum chamber filled with ultrahigh purity Ar gas. Each ingot was remelted in the Cu mold at least five times to improve the macroscopic chemical homogeneity. The ingot was then induction remelted in a vacuum chamber and cast into another water-cooled Cu mold. Each of the resultant ingots was about 25.4 mm (1 inch) wide, 12.7 mm (0.5 inch) thick and 62.2 mm (2.45 inch) long. Their compositions are shown in Table 1. As shown, these alloys are close to but not exactly equimolar in Co, Cr, Fe and Ni. Likewise, the Cu molar ratio x is close to but not exactly 0.5, 1.0, 1.5 or 2.0. For convenience of discussion however, x = 0, 0.5, 1.0, 1.5 and 2.0 will be used to indicate these alloys. 

The ingots were cut by electric discharge machining into small coupons 25.4 mm (1 inch) wide, 1.5 mm (0.06 inch) thick and 62.2 mm (2.45 inch) long. Conventional tests for the susceptibility to solidification cracking, such as the widely used Varestraint test [24] and the recently developed Transverse-Tension weldability test [25], were not used because the coupons were very limited in length and width. Instead, bead-on-plate welding was conducted by gas-tungsten arc welding (GTAW) without a filler metal. GTAW was conducted along the centerline of the coupon along its length direction. The welding current was 50–55 A, the voltage 9.5 V, and the travel speed 1.48 mm/s (3.5 ipm). The polarity was direct current electrode negative. Welding grade Ar was used as the shielding gas, directed at both the top and bottom surfaces of the workpiece at the flow rate of 16.5 L/minute or 35 CFH (cubic feet per hour).

The top surfaces of the resultant welds were examined visually for cracks that occurred during welding. The welds were then cut, polished, and etched electrochemically with a solution consisting of 60 g of oxalic acid in 600 mL of water for microstructure examination by optical microscopy. 

## 3. Results and Discussion

### 3.1. Solidification Cracking

Figure 2 shows an example of an ingot after being cut by electric discharge machining. The as-cut surface is smooth. However, it reveals the internal porosity inside the ingot. 

Figure 3 shows the resultant welds, including the fusion zones, craters (the weld pools that solidified at the end of welding) and surrounding areas. Solidification cracking is known to occur in the fusion zone and/or the weld crater during solidification [26]. Alloy CoCrFeNiCu_0_ shows no cracks anywhere. Alloy CoCrFeNiCu_2_ shows solidification cracks only in the crater. Alloys CoCrFeNiCu_0.5_, CoCrFeNiCu_1_ and CoCrFeNiCu_1.5_ are similar in the sense that they all show solidification cracks in both the fusion zone and the crater and in the sense that centerline cracking propagates throughout the fusion zone. Thus, based on the extent of fusion-zone cracking, alloys CoCrFeNiCu_0.5_, CoCrFeNiCu_1_ and CoCrFeNiCu_1.5_ are most susceptible. It is likely that alloy CoCrFeNiCu_1.5_ could have shown less cracking in the fusion zone than that shown in Figure 3d if cracking were not initiated by the porosity in the base metal near the starting point of the weld. Also, CoCrFeNiCu_0.5_ could have shown much more cracking if it had the same weld length as CoCrFeNiCu_1.5_. Furthermore, in CoCrFeNiCu_0.5_ it is clear that centerline cracking propagates further into the crater.

The extent of cracking in the crater can also be used to evaluate the susceptibility to solidification cracking [27,28]. As compared to the fusion zone, when the arc is turned off suddenly at the end of welding, the weld pool solidifies and shrinks much more rapidly, allowing less time for liquid backfilling from the pool (through intergranular channels) to heal cracks [15]. Thus, the crater can be more sensitive to solidification cracking than the fusion zone. As can be seen in Figure 3, alloys CoCrFeNiCu_0.5_ and CoCrFeNiCu_1_ show the most severe crater cracking in view of their widely open crater cracks. The total length of crater cracks is greater in CoCrFeNiCu_0.5_ than CoCrFeNiCu_1_. Thus, based on cracking both in the fusion zone and the crater, the susceptibility to solidification cracking can be ranked in the decreasing order of CoCrFeNiCu_0.5_ > CoCrFeNiCu_1_ > CoCrFeNiCu_1.5_ > CoCrFeNiCu_2_ > CoCrFeNiCu_0_. It can be said that alloys CoCrFeNiCu0_.5_, CoCrFeNiCu_1_ and CoCrFeNiCu_1.5_ are most susceptible to solidification cracking, followed by CoCrFeNiCu_2_, with CoCrFeNiCu_0_ being the least susceptible.

### 3.2. Solidification Cracking Susceptibility Index

Figure 4 shows the solidification paths of CoCrFeNiCu_x_, that is, the curves of temperature *T* vs. solid fraction *f_S_* during solidification. For alloy CoCrFeNiCu_0_, the freezing temperature range is extremely narrow, only about 16 °C. The primary solidification phase is the fcc phase Fcc1. With Cu as an additional alloying element, however, the freezing temperature range becomes much wider, that is, 298 °C for CoCrFeNiCu_0.5_, 275 °C for CoCrFeNiCu_1_, 264 °C for CoCrFeNiCu_1.5_, and 260 °C for CoCrFeNiCu_2_. The primary solidification phase is still Fcc1 and is Cu-lean. However, as solidification proceeds further, a Cu-rich fcc phase Fcc2 and a Cr-rich bcc phase Bcc also forms [14]. The higher the Cu content of the alloy, the earlier Fcc2 and Bcc start to form from the liquid.

Figure 5 shows the *T-(f_S_)*^1/2^ curves of the alloys. Since the maximum |*dT/d(f_S_)*^1/2^| up to *(f_S_)*^1/2^ = 0.99 is taken as the index for the susceptibility to solidification cracking [18], the curves are shown from *(f_S_)*^1/2^ = 0.80 to *(f_S_)*^1/2^ = 1. As shown by the tangents to the curves up to *(f_S_)*^1/2^ = 0.99, the maximum steepness of the tangent and hence the index decreases in the order of CoCrFeNiCu_0.5_ > CoCrFeNiCu_1_ > CoCrFeNiCu_1.5_ > CoCrFeNiCu_2_ > CoCrFeNiCu_0_. This is consistent with the ranking of the susceptibility to solidification cracking based on the extent of solidification cracking both in the fusion zone and the crater as shown previously. Figure 1 and Figure 5 can help explain the difference between the alloys and their susceptibility to solidification cracking shown in Figure 3. 

### 3.3. Liquation Cracking

Figure 3 also shows liquation cracking can occur in the partially melted zone (PMZ), which is outside but immediately next to the fusion zone, including the fusion boundary (Figure 3b). Liquation cracking tends to initiate in the PMZ at the starting point of welding and near the crater. It often propagates into the fusion zone as solidification cracking. Like solidification cracking, the susceptibility to liquation cracking can be ranked in the decreasing order of CoCrFeNiCu_0.5_ > CoCrFeNiCu_1_ > CoCrFeNiCu_1.5_ > CoCrFeNiCu_2_ > CoCrFeNiCu_0_.

## 4. Micrographs of Welds

### 4.1. Solidification Cracking

Figure 6 shows the microstructure in the fusion zone of alloy CoCrFeNiCu_0_. The dendrites are the Fcc1 phase. The secondary dendrite arms are short and hardly recognizable. There are no secondary phases [13], consistent with the solidification path shown in Figure 4a. The dark dots are pits due to corrosion by the etching solution. As shown in Figure 6b, the solidification grain boundary has moved away from its position at the end of solidification to become a migrated grain boundary (MGB). Due to the high solidification temperature and the absence of a secondary phase at the grain boundary to pin it down, solid-state diffusion has caused grain boundary migration [24]. Figure 7 shows the microstructure in the fusion zone of alloy CoCrFeNiCu_0.5_. Figure 7a shows the microstructure in an area without solidification cracking. The dendrites are the Cu-lean Fcc1 phase. The secondary dendrite arms are short but visible. The Cu-rich secondary phase Fcc2 can be seen in the interdendritic areas. Figure 7b shows the microstructure in an area with solidification cracking. As shown, cracking is intergranular, i.e., along boundaries between dendritic columnar grains. Cu-rich intergranular liquid was present at the moment of cracking.

Figure 8 shows the microstructure in the fusion zone of alloy CoCrFeNiCu_1_. As compared to Figure 7 for alloy CoCrFeNiCu_0.5_, the primary dendrite arms are thinner and the secondary dendrite arms are longer. These more well-developed dendrites suggest increasing constitutional supercooling caused by the higher Cu content in alloy CoCrFeNiCu_1_. Again, cracking is intergranular, and the interdendritic liquid is Cu-rich. The amount of the intergranular liquid is greater in this alloy than that in alloy CoCrFeNiCu_0.5_, consistent with the solidification paths shown in Figure 4b,c. More abundant intergranular liquid and finer dendrites (than those in Figure 7b) are likely to help accommodate transverse tensile strain before cracking occurs. 

Figure 9 shows the microstructure in the fusion zone of alloy CoCrFeNiCu_1.5_. As compared to Figure 8 for alloy CoCrFeNiCu_1_, the dendrites are finer, and the intergranular liquid is wider. Cracking is intergranular and the amount of the Cu-rich intergranular liquid near cracks is greater. 

The microstructure in the fusion zone of alloy CoCrFeNiCu_2_ is shown in Figure 10. Unlike alloy CoCrFeNiCu_1.5_, shown in Figure 9, the dendrites are finer and there is no cracking. It is interesting to note that some dendrites in one grain penetrate into the neighboring grain, as can be seen in Figure 10a and more clearly in Figure 10b. 

Figure 11 shows the microstructure in the fusion zone of alloy CoCrFeNiCu_1_. The image taken by Scanning Electron Microscopy (SEM) shows the interdendritic liquid is Cu-rich (88.5 at% at Point 1) with much more Cu than the Cu-lean Fcc1 dendrites (10.2 at% at Point 2).

### 4.2. Liquation Cracking

An alloy is partially melted when heated up into its melting temperature range, which is the same as the freezing temperature range if the alloy is in the as-cast condition before heating, e.g., the alloys in the present study. During welding of an as-cast alloy, the region immediately outside the fusion boundary that is heated into the freezing temperature range is called the partially melted zone (PMZ) because liquid formation (call liquation) can occur along grain boundaries (and within grains) [26]. Under the tension induced during welding, e.g., by the solidifying and hence contracting mushy zone near the PMZ, cracking can occur along liquated grain boundaries, that is, liquation cracking. Figure 12 shows liquation cracking in the PMZ of alloy CoCrFeNiCu_1_. As shown, liquation cracking is intergranular cracking in the PMZ near the fusion boundary (i.e., the weld edge).

Figure 13a shows the microstructure in alloy CoCrFeNiCu_0_ outside the PMZ, i.e., in the base metal. The thin dark lines are the migrated grain boundaries. The microstructure near the fusion boundary is shown in Figure 13b. The fusion boundary is essentially horizontal. As can be seen, the liquid in the weld pool solidifies first in the planar mode but soon changes to the cellular mode [26]. No liquation cracking is visible, consistent with the absence of cracking in the PMZ near the fusion boundary in Figure 3a. Figure 14a shows the microstructure in alloy CoCrFeNiCu_0.5_ outside the PMZ. Figure 14b shows a liquation crack in the PMZ that propagates into the fusion zone as solidification cracking. In Figure 3b, liquation cracks are visible near the fusion boundary and they propagate into the fusion zone as solidification cracks. 

Similar results can be seen in Figure 15 for alloy CoCrFeNiCu_1_ and in Figure 16 for alloy CoCrFeNiCu_1.5_.

Figure 17a shows the microstructure in alloy CoCrFeNiCu_2_ outside the PMZ. Figure 17b shows no liquation cracking in the PMZ. This is consistent with the absence of liquation cracking near the fusion zone in Figure 3e, where liquation cracks are visible in the PMZ only near the crater.

## 5. Conclusions

(1)Based on cracks observed in both the fusion zone and the crater, the ranking of the susceptibility to solidification cracking in arc welding appears to be CoCrFeNiCu_0.5_ > CoCrFeNiCu_1_ > CoCrFeNiCu_1.5_ > CoCrFeNiCu_2_ > CoCrFeNiCu_0_. It can be said, at least, that alloys CoCrFeNiCu_0.5_, CoCrFeNiCu_1_ and CoCrFeNiCu_1.5_ are most susceptible to solidification cracking, followed by CoCrFeNiCu_2_, with CoCrFeNiCu_0_ being the least susceptible. The same ranking can be shown based on the maximum |*dT/d(f_S_)*^1/2^| up to *(f_S_)*^1/2^ = 0.99 as the index for the susceptibility to solidification cracking.(2)Solidification cracks in the fusion zone often show Cu-rich intergranular liquid near cracks; the higher the Cu content of the alloy, the greater the amount of the liquid.(3)Liquation cracking can occur in the PMZ near the fusion boundary and propagate into the fusion zone as solidification cracking. Similar to solidification cracking, the ranking of the susceptibility to liquation cracking appears to be CoCrFeNiCu_0.5_ > CoCrFeNiCu_1_ > CoCrFeNiCu_1.5_ > CoCrFeNiCu_2_ > CoCrFeNiCu_0_.

## Figures and Tables

**Figure 1 materials-16-05621-f001:**
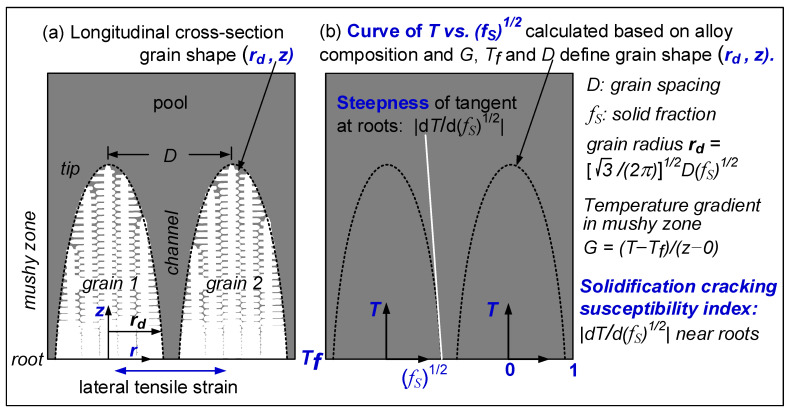
Columnar dendritic grains: (**a**) longitudinal cross-section; (**b**) index for solidification cracking susceptibility proposed by Kou [15].

**Figure 2 materials-16-05621-f002:**
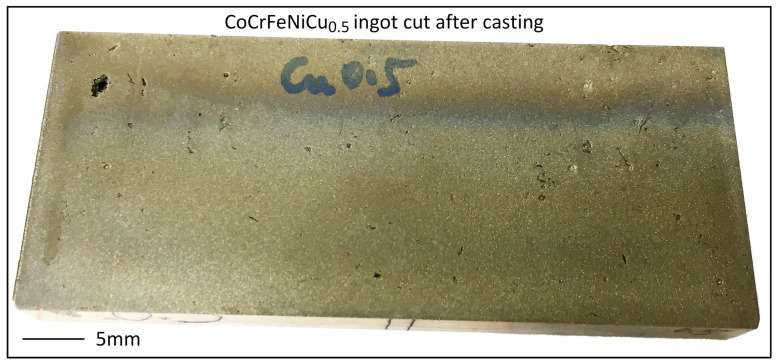
CoCrFeNiCu_0.5_ ingot showing internal porosity exposed after cutting.

**Figure 3 materials-16-05621-f003:**
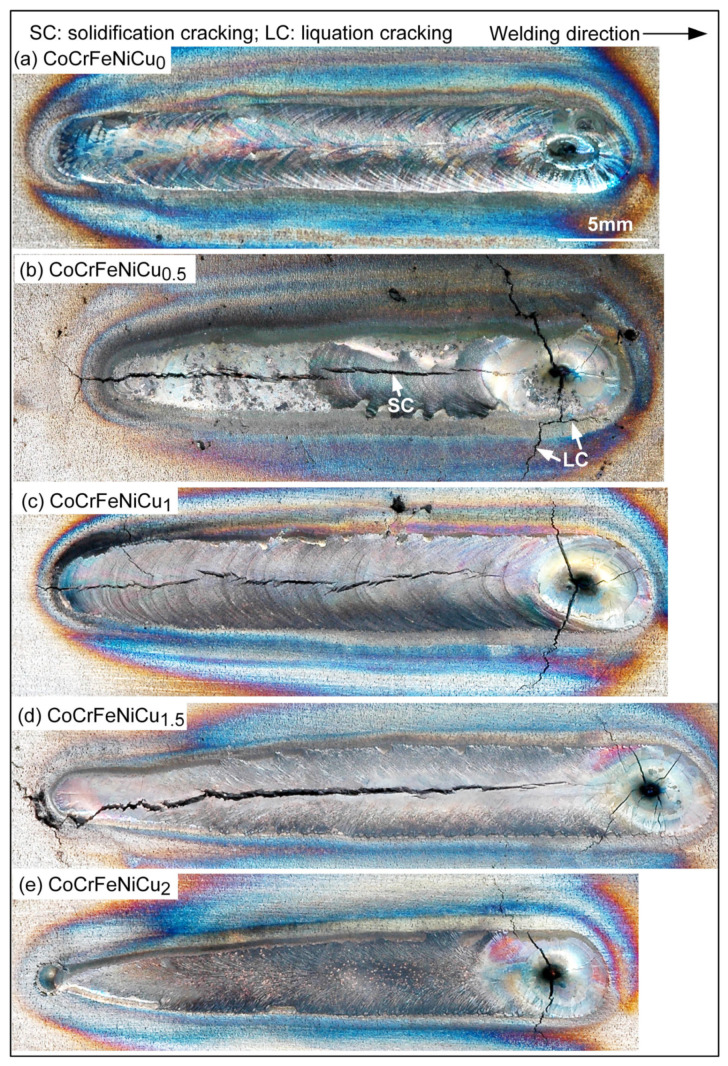
Macrographs showing cracking in welds: (**a**) no cracking anywhere in CoCrFeNiCu_0_; (**b**) through (**d**) cracking in fusion zone (weld metal), crater (end of weld) and partially melted zone (near fusion zone) of CoCrFeNiCu_0.5_, CoCrFeNiCu_1_ and CoCrFeNiCu_1.5_; (**e**) no cracking in fusion zone of CoCrFeNiCu_2_.

**Figure 4 materials-16-05621-f004:**
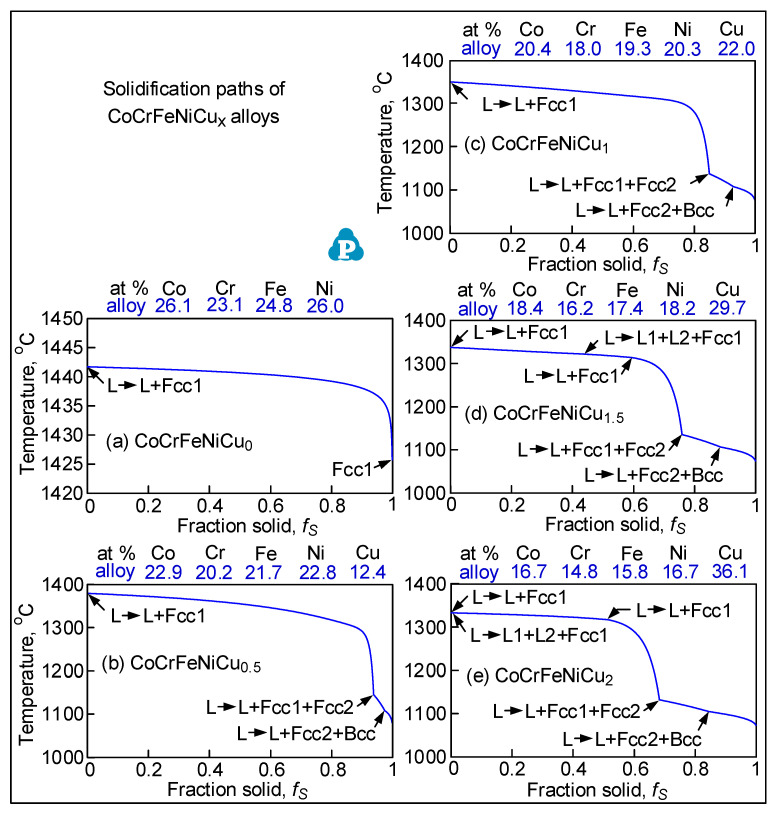
Curves of temperature *T* vs. solid fraction *f_S_* of all alloys during solidification, calculated using commercial thermodynamic software package Pandat2019 [16] (P) and database PanHEA2019 [17] of CompuTherm, LLC, Madison, WI, USA.

**Figure 5 materials-16-05621-f005:**
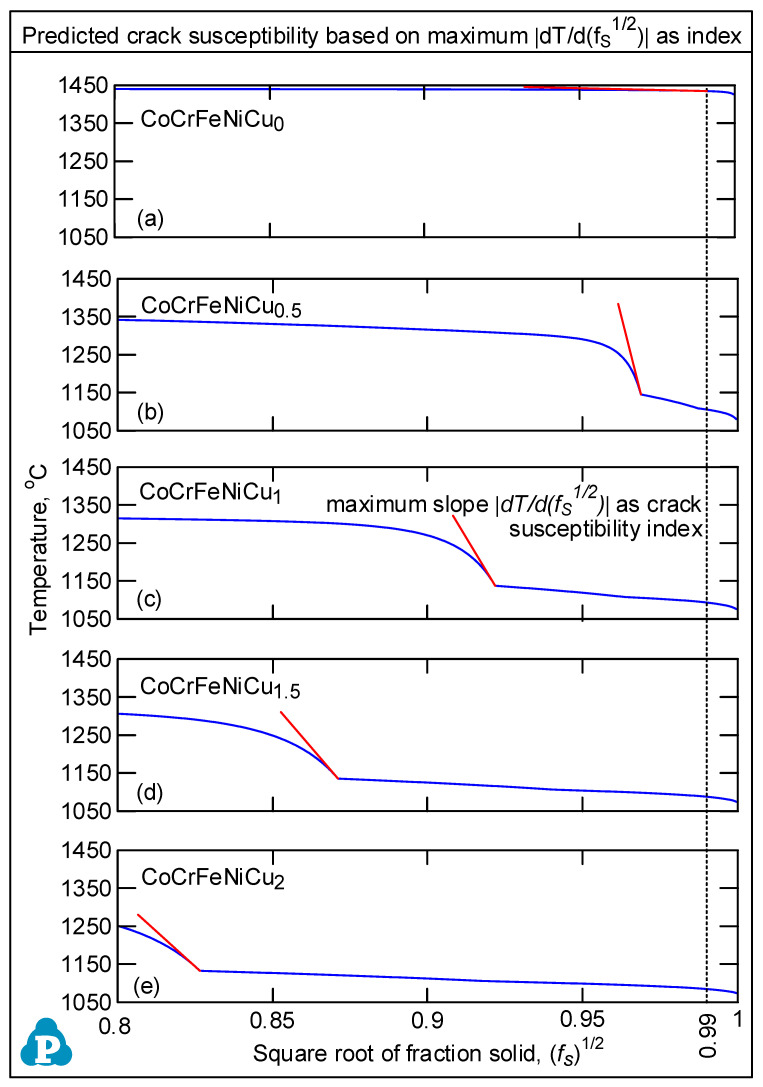
Curves of temperature *T* vs. square root of solid fraction *(f_S_)*^1/2^ of all alloys in the range of 0.8 < *(f_S_)*^1/2^ < 1.

**Figure 6 materials-16-05621-f006:**
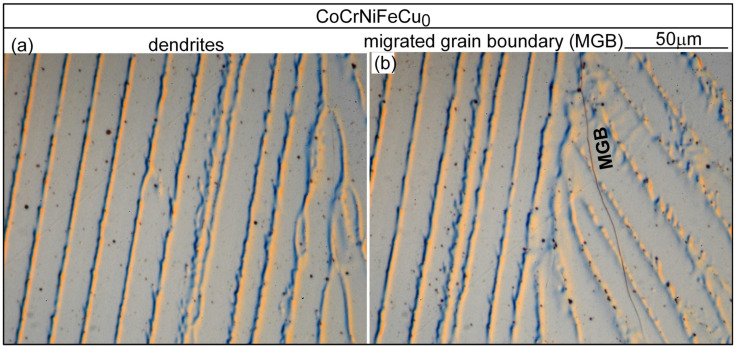
CoCrFeNiCu_0_ fusion zone: (**a**) dendrites; (**b**) migrated grain boundary (MGB).

**Figure 7 materials-16-05621-f007:**
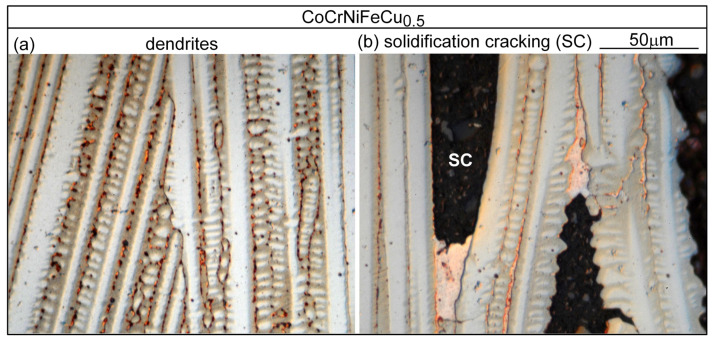
CoCrFeNiCu_0.5_ fusion zone: (**a**) dendrites; (**b**) solidification cracking (SC).

**Figure 8 materials-16-05621-f008:**
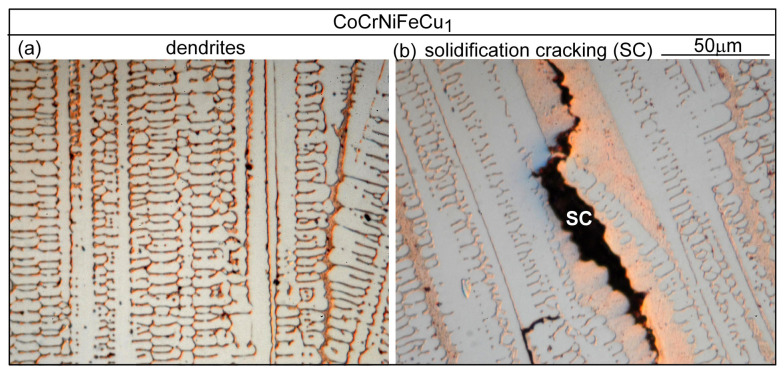
CoCrFeNiCu_1_ fusion zone: (**a**) dendrites; (**b**) solidification cracking.

**Figure 9 materials-16-05621-f009:**
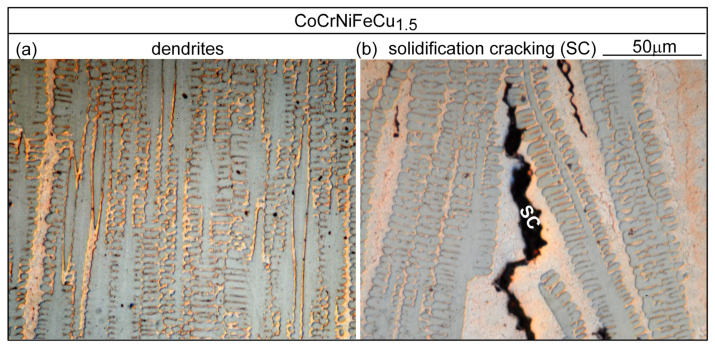
CoCrFeNiCu_1.5_ fusion zone: (**a**) dendrites; (**b**) solidification cracking.

**Figure 10 materials-16-05621-f010:**
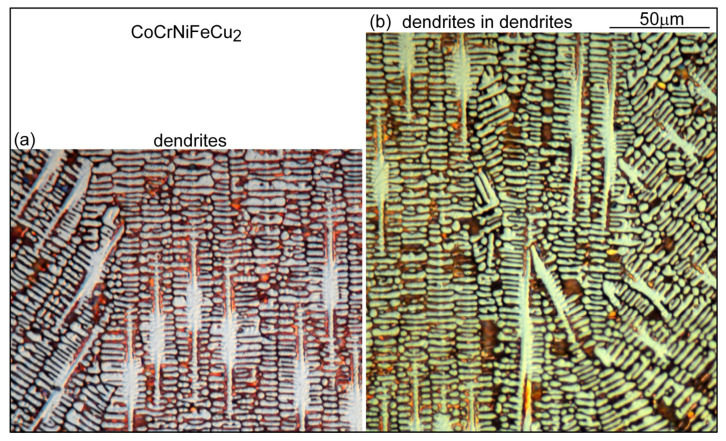
CoCrFeNiCu_2_ fusion zone: (**a**) dendrites; (**b**) dendrites in the right grain penetrating into the left grain.

**Figure 11 materials-16-05621-f011:**
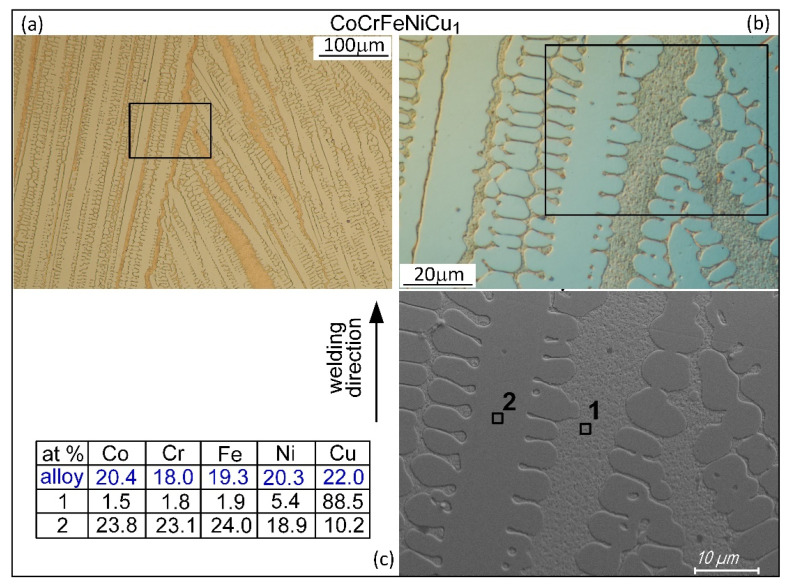
Microstructure in fusion zone of alloy CoCrFeNiCu_1_: (**a**) optical micrograph; (**b**) optical micrograph enlarged; (**c**) SEM image showing Cu-rich interdendritic liquid at Point 1.

**Figure 12 materials-16-05621-f012:**
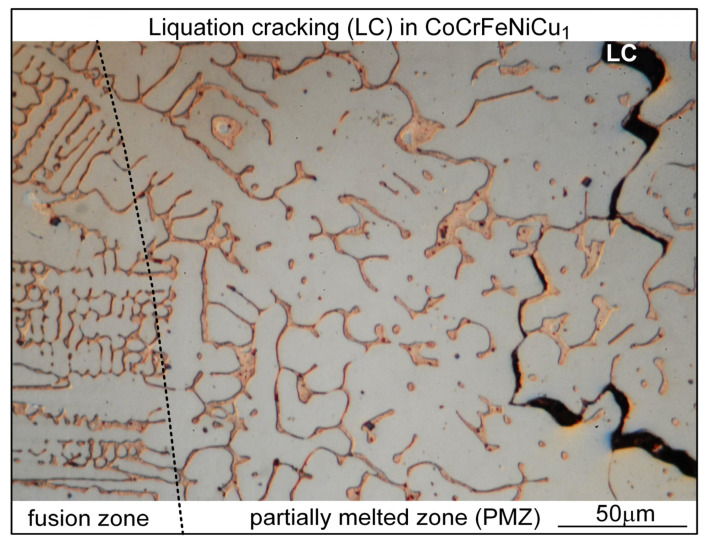
Liquation cracking in alloy CoCrFeNiCu_1_. Dotted line: fusion boundary.

**Figure 13 materials-16-05621-f013:**
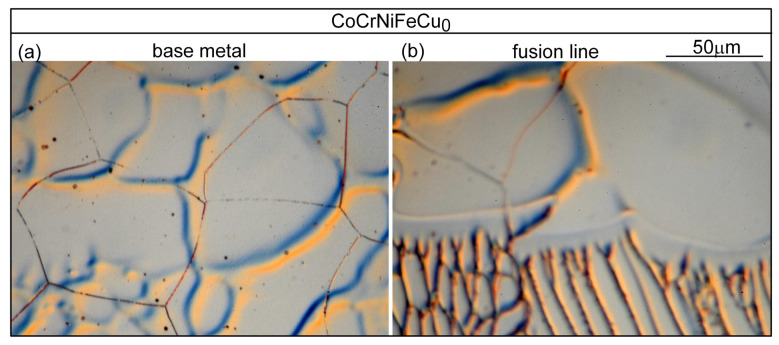
CoCrFeNiCu_0_: (**a**) base metal; (**b**) near fusion boundary. Grain boundary migration is visible.

**Figure 14 materials-16-05621-f014:**
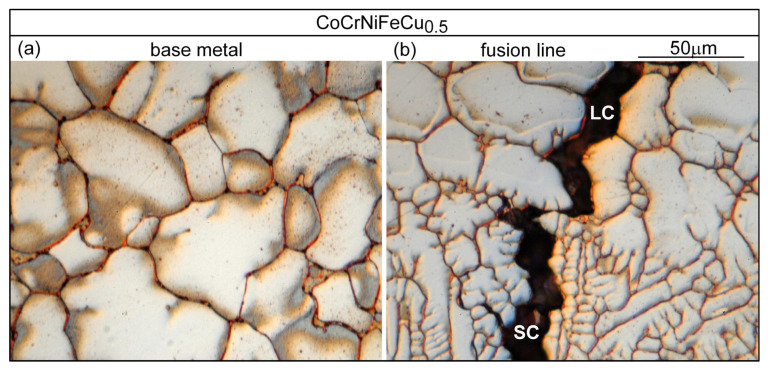
CoCrFeNiCu_0.5_: (**a**) base metal (globular grains, showing slight grain boundary migration); (**b**) near fusion boundary.

**Figure 15 materials-16-05621-f015:**
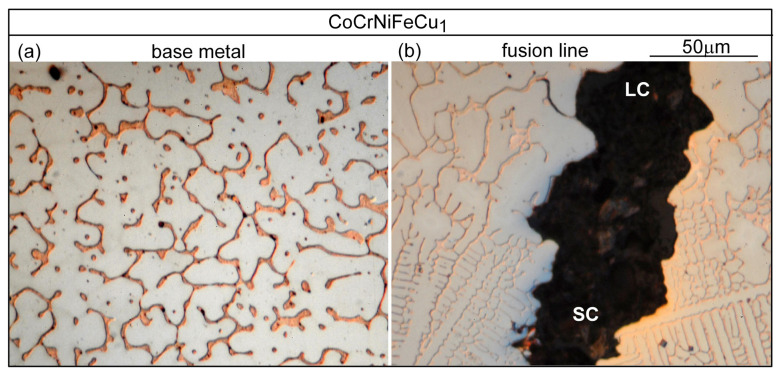
CoCrFeNiCu_1_: (**a**) base metal (dendritic); (**b**) near fusion boundary.

**Figure 16 materials-16-05621-f016:**
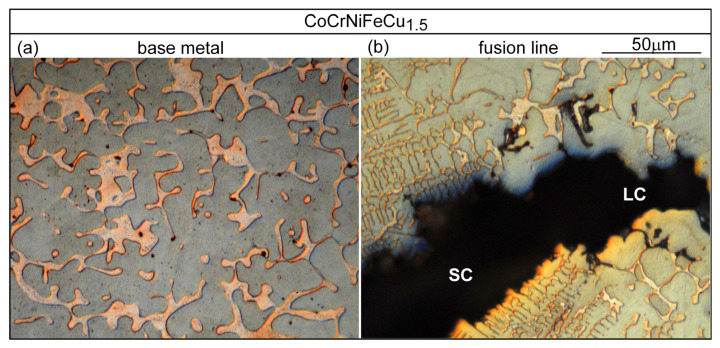
CoCrFeNiCu_1.5_: (**a**) base metal (dendritic); (**b**) near fusion boundary.

**Figure 17 materials-16-05621-f017:**
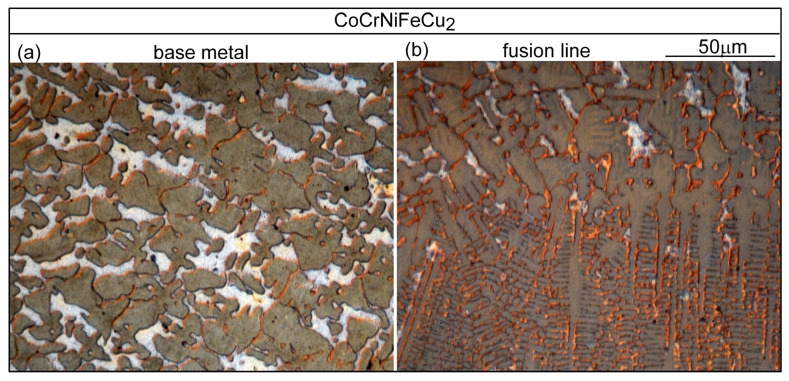
CoCrFeNiCu_2_: (**a**) base metal (dendritic); (**b**) near fusion boundary.

**Table 1 materials-16-05621-t001:** Compositions of CoCrFeNiCu_x_ alloys in atomic%.

Alloy	Co	Cr	Fe	Ni	Cu
CoCrFeNiCu_0_	26.14	23.06	24.77	26.03	0
CoCrFeNiCu_0.5_	22.91	20.21	21.71	22.82	12.35
CoCrFeNiCu_1_	20.39	17.99	19.32	20.31	21.99
CoCrFeNiCu_1.5_	18.37	16.21	17.41	18.23	29.71
CoCrFeNiCu_2_	16.72	14.75	15.84	16.65	36.05

## Data Availability

Not Applicable.

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
