# Peer review of "Solidification and Liquation Cracking in Welds of High Entropy CoCrFeNiCux Alloys"

_materials, 2023, doi:10.3390/ma16165621_

Round 1

Reviewer 1 Report

Dear Authors, for me the whole article is clear and I find it suitable for publication. 

Author Response

Thanks.

Reviewer 2 Report

The paper is sufficiently novel and interesting to warrant publication, it results in an advance to the current scientific literature. The abstract reflect the content and summarize the research problem.

The purpose of the study clearly outlined and the findings of prior works are very deep discussed.

In the experimental part the author accurately explain how the data was collected. The authors give the sufficient information about the preparation of specimens and also sufficient information that the experiment can be reproduced.

The discussion is supported by the results, clearly explained the obtained results focused on the scientific sense. It specifically shows how the work resulted in an advance.

The conclusion sounds and justifiable as based on the results and discussion.

The structure of the paper is well-organized, the information flows in a logical manner.

The English expression is clear, understandable and easy to interpret.

The paper is acceptable in the present form. It gives the adequate reference to the prior work. It does not contain errors in mathematics, logic or experimental procedure. All figures and tables are necessary, they prove their point.

The paper is clearly presented and well organized.

Comments:

page 2 line 80 - CFH - explain

page 5 line 145 - partially meted zone - change to melted

Author Response

1. Thanks.

2. page 2 line 80 - CFH – explain:  CFH means Cubic Feet per Hour as indicated on page 3 line 102.

3. page 5 line 145 - partially meted zone - change to melted: “meted” has been changed to “melted” as shown on page 6 line 180.

Reviewer 3 Report

Introduction: I suggest to improve the discussion about the literature. 

Figure 1: I suggesto to better evidence the internal porosity; for example, with correlated zoom images.

Figure 2: I suggest to better explain the reasons of the different behaviour. Could be interesting to analyse a section of these joints?

Section 3.2: "Fig. 3 shows the solidification... from the liquid". I suggest to better rewrite this sentence improving the description of Figure 3. 

Figure3/4: What is the mark "P" on the figures?

References: I suggest adding more recent references.

Author Response

1. Introduction: I suggest to improve the discussion about the literature.

Response: Discussion on literature has been improved by adding Paragraph 1 and Figure 1 to page 2 (lines 46 through 60).

2. Figure 1: I suggesto to better evidence the internal porosity; for example, with correlated zoomimages.

Response: The image in Fig. 2 has been zoomed in (enlarged) to show porosity more clearly as shown on page 3 lines 113 to 115.

3. Figure 2: I suggest to better explain the reasons of the different behaviour. Could be interesting toanalyse a section of these joints?

Response: With the help of Fig. 1 (new figure), Fig. 5 (original Fig. 4) can explain the difference between the alloys in their susceptibility to solidification cracking shown in Fig. 3 (original Fig. 2).

4. Section 3.2: "Fig. 3 shows the solidification... from the liquid". I suggest to better rewrite thissentence improving the description of Figure 3.

Response: The caption of Fig. 4 (original Fig. 3) has been rewritten as follows: “Fig. 4 Curves of temperature T vs. solid fraction fS of all alloys during solidification, calculated using commercial thermodynamic software package Pandat (P).”

4. Figure3/4: What is the mark "P" on the figures?

Response: “P” is the trademark of “Pandat” of CompuTerm LLC. See the caption of Fig. 4 (original Fig. 3).

5. References: I suggest adding more recent references.

Response: A more recent reference related to both solidification cracking and liquation cracking have been provided on page 12 line314-315 and as Reference 27 on page 13 line 392.
